# Detection of SARS-CoV-2 Using Reverse Transcription Helicase Dependent Amplification and Reverse Transcription Loop-Mediated Amplification Combined with Lateral Flow Assay

**DOI:** 10.3390/biomedicines10092329

**Published:** 2022-09-19

**Authors:** Aleksandra Anna Zasada, Ewa Mosiej, Marta Prygiel, Maciej Polak, Karol Wdowiak, Kamila Formińska, Robert Ziółkowski, Kamil Żukowski, Kasper Marchlewicz, Adam Nowiński, Julia Nowińska, Waldemar Rastawicki, Elżbieta Malinowska

**Affiliations:** 1Department of Sera and Vaccines Evaluation, National Institute of Public Health NIH—National Research Institute, Chocimska 24, 00-791 Warsaw, Poland; 2The Chair of Medical Biotechnology, Faculty of Chemistry, Warsaw University of Technology, 00-664 Warsaw, Poland; 3Centre for Advanced Materials and Technologies CEZAMAT, Warsaw University of Technology, Poleczki 19, 02-822 Warsaw, Poland; 42nd Dept of Respiratory Medicine, National Institute of Tuberculosis and Lung, 01-138 Warsaw, Poland; 5Faculty of Medicine, Medical University of Warsaw, 02-091 Warsaw, Poland; 6Department of Bacteriology and Biocontamination Control, National Institute of Public Health NIH—National Research Institute, 00-791 Warsaw, Poland

**Keywords:** RT-HDA, RT-LAMP, diagnostics, point-of-care, SARS-CoV-2, isothermal amplification, LFA

## Abstract

Rapid and accurate detection and identification of pathogens in clinical samples is essential for all infection diseases. However, in the case of epidemics, it plays a key role not only in the implementation of effective therapy but also in limiting the spread of the epidemic. In this study, we present the application of two nucleic acid isothermal amplification methods—reverse transcription helicase dependent amplification (RT-HDA) and reverse transcription loop-mediated amplification (RT-LAMP)—combined with lateral flow assay as the tools for the rapid detection of SARS-CoV-2, the etiological agent of COVID-19, which caused the ongoing global pandemic. In order to optimize the RT-had, the LOD was 3 genome copies per reaction for amplification conducted for 10–20 min, whereas for RT-LAMP, the LOD was 30–300 genome copies per reaction for a reaction conducted for 40 min. No false-positive results were detected for RT-HDA conducted for 10 to 90 min, but false-positive results occurred when RT-LAMP was conducted for longer than 40 min. We concluded that RT-HDA combined with LFA is more sensitive than RT-LAMP, and it is a good alternative for the development of point-of-care tests for SARS-CoV-2 detection as this method is simple, inexpensive, practical, and does not require qualified personnel to perform the test and interpret its results.

## 1. Introduction

The development of point-of-care (POC) tests for clinical diagnostics has been a subject of studies conducted by many research groups in the world for many years. However, the COVID-19 pandemic has triggered the development of rapid POC tests to address the global urgent need for the simple and rapid detection of SARS-CoV-2—the etiological agent of COVID-19. Due to the rapid spread of the epidemic, a severe course of the disease in many patients, and flu-like or asymptomatic course in others, reliable, easy-to-use, cost-effective, and rapid COVID-19 diagnostic tests, with the potential for non-hospital and non-laboratory use, became critical to counteract the COVID-19 epidemic [1,2].

There are three types of diagnostic tests for COVID-19: tests based on detection of genetic material from the SARS-CoV-2 virus, tests based on the detection of antigens of the virus, and tests that detect specific antibodies produced from immune response to the viral infection. The antigen tests and the antibody (serological) tests are available as a POC format that usually utilizes lateral flow technology. Those tests results can be read in minutes but are relatively insensitive and subject to host immune response limitations [3,4]. The detection of SARS-CoV-2 genetic markers is regarded as the most accurate test but is usually based on the polymerase chain reaction (PCR) and therefore needs to be conducted in a diagnostic laboratory [3,4,5].

Other methods described in scientific literature as potentially useful for COVID-19 diagnostics include CRISPR-Cas-based detection methods, various biosensors and sequencing. However, CRISPR-Cas requires specialized staff for the RNA extraction step and is less sensitive than RT-PCR. Biosensors targeting antibodies, viral proteins, or nucleic acids are promising tools for COVID-19 diagnostics, but most of the developed biosensors are still in the process of optimization and validation. Sequencing techniques enable the accurate identification of SARS-CoV-2 RNA and are not affected by the emergence of new variants of the virus. However, sequencing is not recommended for large-scale testing in emergency situations because it is an expensive and slow approach, and very experienced personnel are needed to perform the test and interpret the results. However, the application of sequencing is crucial for the detection new mutations and new variants of SARS-CoV-2 [6,7].

In the study, we present the application of two nucleic acid isothermal amplification methods (reverse transcription helicase dependent amplification (RT-HDA) and reverse transcription loop-mediated amplification (RT-LAMP)), combined with lateral flow assay, as the tools for rapid detection of SARS-CoV-2 genetic markers that might be applied in point-of-care testing.

## 2. Materials and Methods

### 2.1. Nucleic Acids Template for Amplification

The synthetic target sequences for gene E, gene RdRP, and gene N of SARS-CoV-2 (Table 1) were designed based on the sequence of SARS-CoV-2 isolate Wuhan-Hu-1, a complete genome, available in GenBank under the accession no. MN908947.3 and ordered at Bio-Synthesis Inc. (Lewisville, Texas, USA). The purified RNA from cultivated SARS-CoV-2 clinical isolates—AMPLIRUN SARS-CoV-2 RNA control—was purchased from Vircell S.L. (Granada, Spain).

### 2.2. Clinical Samples and RNA Extraction

The nasopharyngeal swabs were collected from patients with suspected COVID-19. The study was approved by the Bioethics Committee of the National Institute of Public Health—the National Institute of Hygiene (NIPH-NIH). Each participant gave their written consent to the study. The samples were collected using the Virus RNA Collection Kit (Xiamen Zeesan Biotech Co., Ltd., Xiamen, China). The RNA was extracted from 180 µL of each sample. The extraction was conducted using the MGI Magnetic Beads Virus RNA/DNA Extraction Kit (Wuhan MGI Tech Co., Ltd., Wuhan, China) and the MGISP-960 apparatus, according to the manufacturer’s instructions. The presence of SARS-CoV-2 RNA in the samples was verified with the RT-PCR assay using the Real-Time Fluorescent RT-PCR Kit for Detecting SARS-CoV-2 (BGI, Shenzhen, China) according to the manufacturer’s instruction. Briefly, 10 μL RNA was added to 20 μL PCR mix. Amplification was conducted using Applied Biosystems 750 thermocycler (Applied Biosystems, Waltham, MA, USA) with reverse transcription at 50 °C for 20 min, a cDNA initial denaturation step at 95 °C for 10 min, and 40 cycles at 95 °C for 15 s and 60 °C for 30 s. A sample was regarded as positive when the cycle threshold (Ct) value of the target *ORF1ab* region was ≤38 (≤32 for highly positive sample).

### 2.3. Oligonucleotide Primers

We used oligonucleotide primers recommended by the WHO and available in the scientific literature as well as designed in this study with the use of the PrimerQuest tool (https://eu.idtdna.com/PrimerQuest/Home/Details/0_4, accessed on 18 September 2022). The primers details are presented in Table 2. The specificity of designed primers was verified in silico using the GenBank database. Unmodified primers were ordered from Genomed (Warsaw, Poland). Primers modified with biotin and fluorescein isothiocyanate (FITC) were ordered form Metabion (Monachium, Germany).

### 2.4. Reverse Transcription and PCR

At the first step, oligonucleotide primers pairs for RT-HDA were verified by the RT-PCR using the synthetic target sequences and RNA extracted from clinical samples. Reverse transcription was conducted using the TranScriba Kit (A&A Biotechnology, Gdansk, Poland) with primer oligo(dT)_18_, according to the manufacturer’s instruction. The reaction was conducted for 60 min at 42 °C. Then, 2 µL of obtained cDNA was used for the PCR. The PCR reaction mixture contained 10 µL of NXT Taq PCR Master Mix (EURX, Gdansk, Poland), 0.5 µM of each forward and reverse primers, and 2 µL of the template and was carried out in a final volume of 20 µL. The reaction conditions were as follows: initial denaturation at 95 °C for 3 min, 35 cycles of denaturation at 96 °C for 5 s, annealing at 58 °C for 5 min, and elongation at 68 °C for 5 s. The final cycle was followed by extension at 72 °C for 1 min. In total, 10 μL of the PCR product were loaded on 2% agarose gel. The size of the amplicon was verified by comparison with the molecular DNA marker Perfect 100 bp Ladder (EURX, Gdansk, Poland).

### 2.5. RT-HDA Reactions

RT-HDA was performed using the IsoAmp II Universal tHDA Kit (New England Biolabs, Ipswich, MA, USA) and ProtoScript II Reverse Transcriptase (New England Biolabs, Ipswich, MA, USA), according to the manufacturer’s protocols. Briefly, reverse transcription was conducted with gene-specific primers at 42 °C for 30 min. Then, the HDA reaction was carried out in a final volume of 50 μL. The reaction mixture contained 75 nM of each forward and reverse primer; 3.5 mM MgSO_4_; 20 mM NaCl; Annealing Buffer II; IsoAmp dNTP solution; IsoAmp Enzyme Mix; and 2 μL cDNA. In the negative control, molecular grade water was used instead of a sample template. The reaction was conducted at 65 °C for 90 min.

### 2.6. RT-LAMP Reactions

Each RT-LAMP primer set included two outer (F3, B3), two inner (FIB, BIP), and two loop primers (LF, LB). The FIB and BIP primers were labeled with biotin and FITC, respectively, at the 5′ end to enable the detection of the amplification product with lateral flow strips. The RT-LAMP was carried out in a final reaction volume of 25 μL. The RT-LAMP reaction mixture contained 0.8 μM of FIB and BIP primers each, 0.2 μM of F3 and B3 primers each, 0.4 μM of LF and LB primers each, 1× reaction buffer containing 20 mM Tris-HCl, 50 mM KCl, 10 mM (NH_4_)_2_SO_4_, 2 mM MgSO_4_, 0.1% Tween 20 (New England Biolabs, Ipswich, MA, USA), 0.2 mM dNTP (Sigma-Aldrich, St. Louis, MI, USA), 0.2 M betaine (Sigma-Aldrich, St. Louis, MI, USA), 8 units of Bst 2.0 WarmStart DNA Polymerase (New England Biolabs, Ipswich, MA, USA), 7.5 units of WarmStart RTx Reverse Transcriptase (New England Biolabs, Ipswich, MA, USA), and 2 μL template. In negative control, molecular grade water was used instead of a sample template. The reaction was conducted at 65 °C for 60 min.

### 2.7. Lateral Flow Assay

The results of amplification using RT-HDA and RT-LAMP were read with the use of lateral flow HybriDetect strips (Milenia Biotec, Gießen, Germany). Ten microliters of the sample was pipetted directly on the sample application area of the strip, and the strip was placed into 100 μL of HybriDetect Assay Buffer. The results were read within a few minutes, when a control band became visible (usually after 1–3 min). The results were regarded as positive when two bands were visible (a control band and a test band) or as negative when only the control band was visible. If the control band was not visible up to 15 min of incubation, the test was regarded as invalid.

### 2.8. Determination of the Detection Limit and the Minimal Reaction Time

The limits of detection (LOD) of RT-LAMP and RT-HDA were assessed by way of 10-fold dilutions of AMPLIRUN SARS-CoV-2 RNA control (Vircell S.L., Granada, Spain). Additionally, the LOD of RT-HDA was determined by 10-fold dilutions of a synthetic target sequence.

To determine the required minimal RT-LAMP reaction time, we examined the results of the reactions after 20, 30, 40, 50, and 60 min of incubation using 10-fold serial dilutions of the AMPLIRUN SARS-CoV-2 RNA control (Vircell S.L., Granada, Spain).

To determine the required minimal RT-HDA reaction time, we examined the results of the reactions after 10, 20, 30, 45, 60, 75, and 90 min of incubation using 10-fold serial dilutions of the synthetic template and the AMPLIRUN SARS-CoV-2 RNA control.

### 2.9. Determination of Specificity, Sensitivity, Accuracy, and Reproducibility

The specificity and sensitivity of RT-LAMP and RT-HDA were investigated using 50 clinical samples (25 positive and 25 negative) and RT-PCR as the gold standard. The sensitivity was calculated as follows: A/(A + C) × 100%, and the specificity was calculated as follows: D/(B + D) × 100%, where A is the number of true positive results, B is the number of false-positive results, C is the number of false-negative results, and D is the number of true negative results. Additionally, positive (PPV) and negative predictive value (NPV) were calculated as follows: PPV = A/(A + B) × 100%, NPV = D/(C + D) × 100%. The accuracy of the test was calculated as (A + D)/(A + B + C + D) × 100%.

Reproducibility was investigated using the AMPLIRUN SARS-CoV-2 RNA control (Vircell S.L., Granada, Spain) and clinical samples. All of the tests were performed by at least two laboratory workers and repeated at least three times.

## 3. Results

### 3.1. Verification of Oligonucleotide Primers Using PCR

All of the primer pairs selected for RT-HDA were initially verified with a conventional PCR using synthetic target sequences as a template and RNA extracted from positive clinical samples. In PCR, we obtained amplified products of a proper size for all the primers pair except for the Gen_N-90F/R primer pair, and therefore the said primer pair was excluded from the further study (Figure 1). The PCR results were also checked on the lateral flow strips. All of the positive samples gave two bands on the strips. No false-positive results, caused by non-specific primer interactions, were observed (Figure 2).

### 3.2. RT-HDA

The RT-HDA assay was optimized with a series of experiments with varied concentrations of primers (50–200 nM) and varied concentrations of MgSO_4_ (3–4.5 mM) and HCl (20–50 mM). First, the experiments were conducted using a synthetic template and then verified using RNA extracted from clinical samples. The final optimal reaction mixture for primer pairs E_Sarbeco_F/R, nCoV_IP4-14059Fw/14146Rv, 2019-nCoV_N2-F/R, and Gen_E-107 was the one described in the Materials and Methods section. For the Gen_N-105F/R primer pair, we did not obtain the amplification product in RT-HDA. The amplified products obtained with the aforementioned primer pairs were of the expected size verified using agarose gel electrophoresis.

We observed an unexpected LOD for RT-HDA as we saw a weak test band on the lateral flow strips for the reaction conducted for 90 and 75 min with 0.0002 µM of the synthetic template as well as for the reaction conducted for 20 and 10 min with 0.02 µM of the template. Strong test bands were visible in the case of a reaction conducted for at least 10 min with 0.2 µM of the synthetic template and at least 45 min for 0.02 µM of the template (Table 3, Figure 3). Using the AMPLIRUN SARS-CoV-2 RNA control as a template, we were able to detect 3 copies/reactions. A weak band was visible on the lateral flow strip when the amplification was conducted for 10 min, and a strong band was visible when the amplification was conducted for 20 min (Table 4, Figure 4). We did not detect any false-positive results.

The sensitivity and specificity of the RT-HDA were calculated as 100%. PPV, NPV, and accuracy were also 100%. The results of the test performed by two laboratory workers in triplicate were comparable.

### 3.3. RT-LAMP

The RT-LAMP assay was optimized with a series of experiments with varied concentrations of primers and varied concentrations of MgSO_4_ (3–8 mM). The reaction was optimized as a multiplex reaction for the simultaneous detection of two markers: gene E and gene N. The experiments were conducted using RNA extracted from clinical samples. The final optimal reaction mixture has been described in the Materials and Methods section.

The LOD for RT-LAMP combined with LFA was 3000 copies/reactions for 30 min of incubation. We obtained a weak band for 30 copies/reactions and a strong band for 300 copies/reactions when the RT-LAMP was conducted for 40 min. However, we also noticed strong false-positive results when the incubation time was 50 and 60 min (Table 5, Figure 5).

The sensitivity and specificity of the RT-LAMP were calculated as 100% and 83.3%, respectively. PPV and NPV were 100%, and accuracy was 91%. The results of the test performed by two laboratory workers in triplicate were comparable.

We verified the specificity of the RT-LAMP conducted for 40 min using clinical samples. We obtained positive results for the positive clinical samples tested in agarose gel electrophoresis and on lateral flow strips when the RT-LAMP reaction time was 40 min (Figure 6). We did not observe false-positive results for the negative control incubated for 40 min.

## 4. Discussion

The rapid and accurate detection and identification of pathogens in clinical samples is essential for all infection diseases since it enables appropriate and early treatment, which increases the chances for the patient’s full recovery and/or minimizes the sequelae of the infection. However, in the case of epidemics, especially airborne diseases, the rapid and accurate detection of the pathogen is crucial in the scope of limiting the spread of the epidemic.

The application of molecular biology methods, such as the PCR, to detect pathogen-specific fragments of nucleic acid (genetic markers) in microbiological diagnostics significantly accelerated the diagnosis of infections compared to conventional microbiological methods. PCR tests are highly sensitive and specific but need precise thermocycling, and therefore the tests need to be conducted at a laboratory facilitated with appropriate equipment, which is difficult to miniaturize. The development of real-time PCR has enabled the faster detection of pathogens but made the testing even more dependent on the laboratory because of the more complex instruments needed.

Before the COVID-19 pandemic, diagnostics based on the detection of genetic markers were performed mostly with conventional PCR, real-time PCR, and reverse-transcriptase PCR (RT-PCR) assays. The COVID-19 pandemic has created, for the first time, a situation where rapid nucleic acid amplification tests have been needed at such a large scale. A response to those needs is the application of isothermal amplification methods that do not need thermocycling and have the potential to be applied in point-of-care tests. Isothermal amplification methods started to emerge within a few years after the invention of the PCR [11], but they have flourished as a diagnostic tool in the last few years.

We propose the application of two isothermal amplification methods: RT-LAMP and RT-HDA combined with LFA for the rapid detection of SARS-CoV-2, with the potential to be applied in point-of-care testing. Among various isothermal amplification methods, the RT-LAMP is the most widely used for the development of COVID-19 rapid tests e.g., [10,12,13,14,15,16]. The HDA assay is uncommon in the detection of SARS-CoV-2, but it was successfully applied for the detection of other viruses, bacteria, and parasites, e.g., [17,18,19,20]. This method was also applied to analyze nucleic acids from various clinical samples, including blood [16], plasma [21], urine [22], stool [23], and vaginal swabs [24].

As the first response to the need of rapid POC SARS-CoV-2 tests, the antigen and antibody LFA tests were developed. Although they are easy to perform, the tests have major limitations, which have been overcome by LFA combined with nucleic acid amplification tests. For example, the antibody and antigen LFA tests rely on the detection of only one SARS-CoV-2 protein, which may lead to false negatives, especially when some mutations occurs [7]. The RT-LAMP and RT-HDA developed in our study have a multiplex format, which reduces the risk of false negatives because even if a mutation occurs in one target gene the second target gene can still be detected. Additionally, the sensitivity of RT-HDA and RT-LAMP is higher and has been estimated 100% for both methods in our study, whereas the reported sensitivity of antigen tests is 56.2% with 27.9% of false negatives [7]. Table 6 presents a summary of LFA-based tests analyzed by Zhang et al. [2].

The nucleic acid amplification methods can be combined with colorimetric detection, which enables one to read results directly in a test tube by the color change. Various color indicators might be used, such as pH indicators, ds-DNA binding dyes, and metal ions indicators [10,25]. However, the color change might be disturbed by various factors, for example, substances present in an unpurified sample or the pH of a sample. Moreover, a color change read by the eye might be problematic, especially when the color change is partial. The application of LFA ensures that the interpretation of results is easy: the test band is fully visible, and the control band is able to verify if the results are not disturbed by unknown factors.

As we concluded in our previous study [17], HDA has limited application value due to the fact that only relatively small fragments of nucleic acids are amplified. This is because of the limited speed and processivity of the helicases in the reaction [26]. However, in the case of COVID-19 diagnostics, small RNA fragments are recommended for SARS-CoV-2 detection. Therefore, HDA seems to be the perfect tool for the development of POC nucleic acid amplification tests to detect SARS-CoV-2. It is believed that HDA is more restrictive regarding primer design and amplicon selection than PCR [27,28]. However, in accordance with our previous experience, as well as other researchers’ observations [27], a primer pair used in the PCR very often works well also in HDA. Therefore, in our study, we used primers recommended by the WHO for the PCR based COVID-19 diagnostics and new primers designed in this study we verified in the PCR. Among the three primer pairs designed in this study, only one proved to be useful for HDA assay, but all the three primers pairs recommended by the WHO for the PCR detection of SARS-CoV-2 also proved to be suitable for HDA. The advantage of the application of primers recommended by the WHO is the availability of information concerning the well-verified specificity of the primers and possible alerts when new variants undetectable by these primers emerged. Wang et al. [3] analyzed mutations on the primers frequently utilized in diagnostic tests using SARS-CoV-2 genome sequences deposited in the GISAID database. A small number of point mutations were identified on the primers E_Sarbeco_F, E_Sorbeco_R, nCoV_IP4-14059Fw, and nCoV_IP4-14146Rv, which makes the primers still useful for diagnostic purposes. Lopez-Rincon et al. [29] investigated the National Genomics Data Center repository, GISAID, and the National Center for Biotechnology Information and Global Initiative on Sharing All Influenza Data repositories using deep learning coupled with explainable artificial intelligence techniques and revealed that the frequency of the appearance of the primer 2019-nCoV_N2-R sequence (also called US-CDC-N2-R) among samples of the GISAID dataset was 99.74% and that the primer was highly specific. The frequency of the appearance of the primer 2019-nCoV_N2-F sequence (also called US-CDC-N2-F) is 99.43%, but it is not specific to SARS-CoV-2 as it binds also to SARS-CoV.

The limit of detection for RT-HDA combined with LFA established in the study depends on the incubation time. We concluded that the optimal incubation time for the RT-HDA reaction was 20 min. However, when the template concentration is high, positive results can be read after only 10 min of incubation. The LOD for HDA in our study was three copies/reactions and 0.002 µM/reaction in the case of the synthetic template, which is similar to the LOD for commercial RT-PCR tests for SARS-CoV-2. Some researchers point at the nonspecific amplification phenomenon due to template-independent primer interactions. The problem was observed for the detection of low copy number targets due to the long time required for the amplification [19,30]. We did not face the problem in our study as even after 90 min of incubation we did not observe any test band on the immunochromatographic strip for the negative control. However, we noticed this phenomenon in the RT-LAMP assay.

For the RT-LAMP assay, we used a set of primers described by Zhang et al. [10], which were applied for the commercially available test kit for the colorimetric detection of SARS-CoV-2. We used the primers set for gene E and gene N in a duplex reaction to increase the detection frequency [10,14,31,32,33]. The LOD of RT-LAMP combined with LFA was similar to those obtained by Zhang et al. [10] and comparable to the RT-HDA test developed in this study. The use of the duplex format did not influence the LOD compared to the reaction conducted with the primer set for a single gene, but in the case of mutation in a single gene the test in the duplex format still will be useful for the detection of the virus. We observed false-negative results in the RT-LAMP negative control when the incubation time was longer than 40 min. This phenomenon is probably related to primer interactions, particularly the primers labeled with biotin and FITC because the lateral flow strips used in this study detect only fragments labeled with biotin and FITC at both ends. Zhang et al. [10] used the same primer cocktail in the colorimetric RT-LAMP reaction and did not report any false-positive results in the negative control, but the reaction was conducted for a maximum period of 40 min. RT-LAMP false-positive results were reported by other researchers [15]. Shortening the incubation time might lead to negative results for low copy number targets, but the selection of a more sensitive amplicon detection method might suppress this limitation. The lateral flow strips are 100–1000 times more sensitive than agarose gel [2,33]. To enable the detection of RT-LAMP reaction products, we labeled loop primers to obtain the highest possible sensitivity because, according to the mechanism of LAMP, the most amplicons are created from this primer pair. Jaroenram et al. [34] combined LAMP with lateral flow dipsticks, but they used an additional labeled DNA probe that was added to the reaction mixture once the amplification was completed. In our study, we labeled primers normally used for the reaction to speed up the test and minimize the number of pipetting steps and the risk of contamination as each pipetting step poses such a risk, especially when the test is performed outside a laboratory.

A lateral flow assay (also called a lateral flow immunoassay or an immunochromatographic test) has been applied in various COVID-19 tests for the rapid detection of SARS-CoV-2 antigens directly from a clinical sample as well as for the detection of SARS-CoV-2 antibodies [35]. However, the antigenic and serological test based on LFA suffers from low sensitivity [36,37,38]. The gold standard for COVID-19 diagnostics is the detection of the nucleic acid of the virus. The application of labeled primers and antibodies against the labels enabled the application of LFA technology for the detection of amplified fragments of nucleic acids. The combination of LFA with nucleic acid amplification methods makes the rapid COVID-19 test more sensitive.

It must be kept in mind that three steps of the nucleic acid amplification test influence the sensitivity: the extraction of nucleic acids, and the amplification reaction and detection of the amplicons. The type of sample tested and the method of its collection also have a significant impact on the test results. For example, Thi et al. [13] used three different samples for RT-LAMP: a clinical swab, a heated clinical swab, and extracted RNA and revealed that the best results of amplification were obtained with extracted RNA. Kundrod et al. [12] tested nasopharyngeal swab eluates, nasal swab eluates, and saliva samples and showed that the limit of detection is higher for saliva samples. In our previous work, we demonstrated that the type of swab used for the sampling might strongly influence the results of diagnostic tests [39]. Nevertheless, those aspects were not investigated in this study.

## 5. Conclusions

RT-HDA and RT-LAMP combined with LFA developed in our study are reliable and cost-effective tests with the potential to be used in POC testing. The combination of RT-LAMP with LFA is another option to apply this method for COVID-19 diagnostics. However, RT-HDA combined with LFA is more sensitive than RT-LAMP and constitutes a good alternative for the development of POC tests for SARS-CoV-2 detection as this method is simple, inexpensive, practical, and does not require qualified personnel for the performance of the test or the interpretation of its results. RT-HDA is conducted with only one primer pair per target, which minimizes the risk of false-positive results due to primer interactions compared to RT-LAMP, where a mixture of three primers pairs per target are used. Moreover, the use of primer pairs recommended by the WHO for RT-HDA is also advantageous, due to their suitability for the diagnosis of emerging new variants of the virus; the use of these primers is monitored.

## Figures and Tables

**Figure 1 biomedicines-10-02329-f001:**
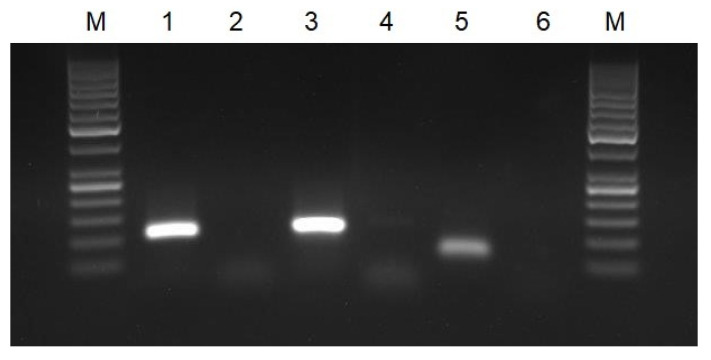
Agarose gel electrophoresis of PCR products: M—molecular DNA marker 100 bp; 1—PCR for gene RdRP (primers nCoV_IP4-14059Fw/Rv); 2—negative control for gene RdRP; 3—PCR for gene E (primers E_Sarbeco_F/R); 4—negative control for gene E; 5—PCR for gene N (primers 2019-nCoV_N2-F/R); and 6—negative control for gene N.

**Figure 2 biomedicines-10-02329-f002:**
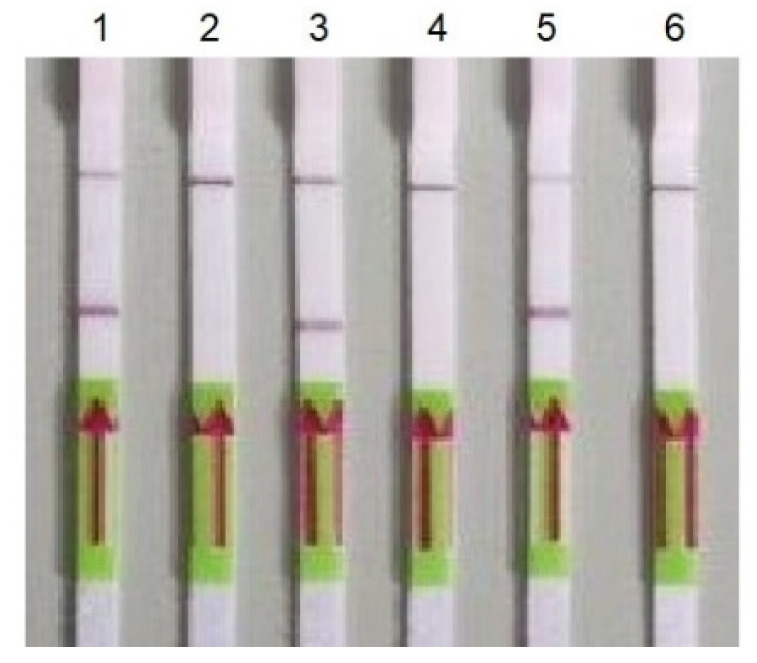
Detection of PCR products using LFA. 1—gene RdRP (primers nCoV_IP4-14059Fw/Rv); 2—negative control for gene RdRP; 3—gene E (primers E_Sarbeco_F/); 4—negative control for gene E; 5—gene N (primers 2019-nCoV_N2-F/R); and 6—negative control for gene E.

**Figure 3 biomedicines-10-02329-f003:**
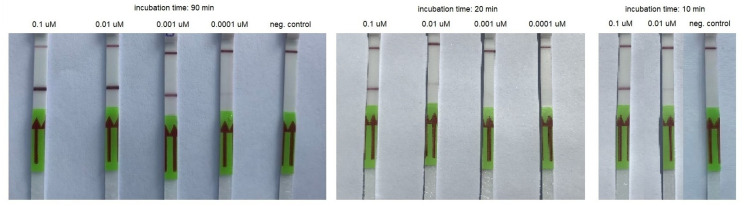
An example of RT-HDA combined with LFA performed using serial dilutions of the synthetic template and various incubation times.

**Figure 4 biomedicines-10-02329-f004:**
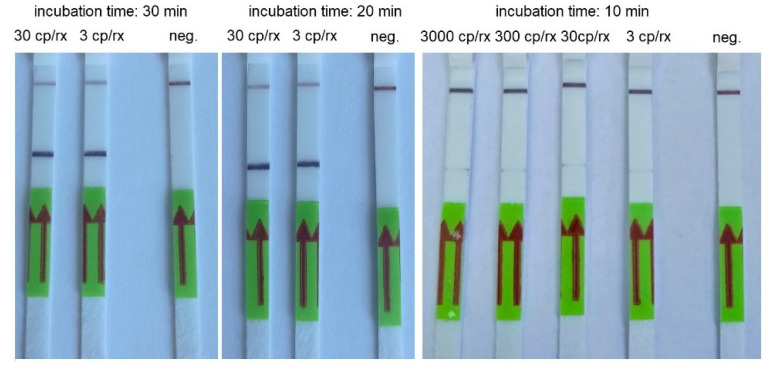
An example of RT-HDA combined with LFA performed using serial dilutions of AMPLIRUN SARS-CoV-2 RNA control and various incubation times. cp/rx—copies/reactions; neg.—negative control.

**Figure 5 biomedicines-10-02329-f005:**
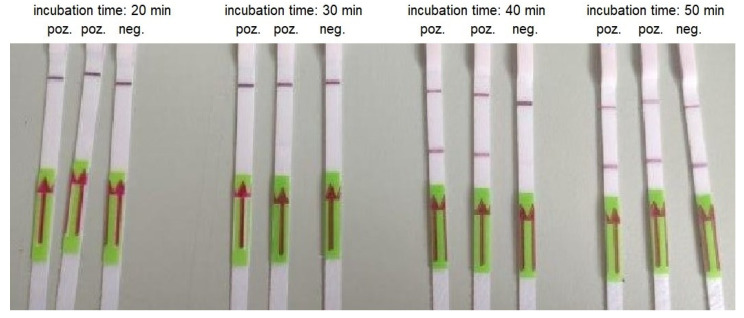
An example of RT-LAMP combined with LFA performed using AMPLIRUN SARS-CoV-2 RNA control at concentration 300 copies/reactions and various incubation times.

**Figure 6 biomedicines-10-02329-f006:**
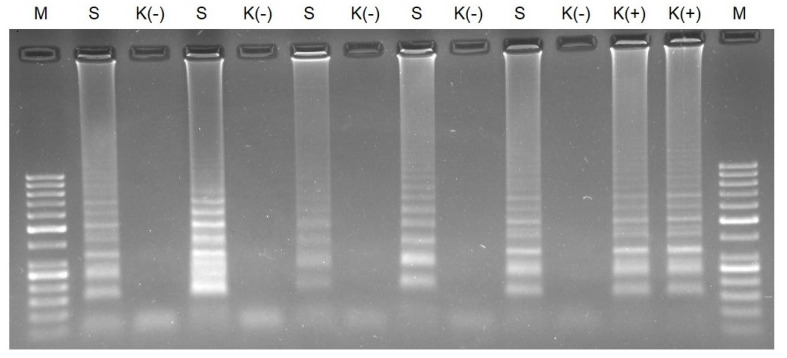
Agarose gel electrophoresis of RT-LAMP for clinical samples conducted for 40 min. M—molecular DNA marker 100 bp; S—clinical sample; K (-)—negative control; and K (+)—AMPLIRUN SARS-CoV-2 RNA control at concentration 3000 copies/reactions.

**Table 1 biomedicines-10-02329-t001:** Sequences of SARS-CoV-2 used as targets in RT-HDA assay.

Target	Oligonucleotide Sequence	Size
Gene E	ACAGGTACGTTAATAGTTAATAGCGTACTTCTTTTTCTTGCTTTCGTGGTATTCTTGCTAGTTACACTAGCCATCCTTACTGCGCTTCGATTGTGTGCGTACTGCTGCAATAT	113 bp
Gene RdRP	GGTAACTGGTATGATTTCGGTGATTTCATACAAACCACGCCAGGTAGTGGAGTTCCTGTTGTAGATTCTTATT ATTCATTGTTAATGCCTATATTAACCTTGACCAG	107 bp
Gene N	TTACAAACATTGGCCGCAAATTGCACAATTTGCCCCCAGTTCTTCGGAATGTCGCGC	57 bp

**Table 2 biomedicines-10-02329-t002:** Primer sets used in the study.

Isothermal Amplification Assay	Primer Name	Primer Sequence	Target Gene of SARS-CoV-2	Reference
RT-HDA	E_Sarbeco_F	ACAGGTACGTTAATAGTTAATAGCGT	Gene E	[8]
	E_Sarbeco_R	ATATTGCAGCAGTACGCACACA		
	nCoV_IP4-14059Fw	GGTAACTGGTATGATTTCG	Gene RdRP	[9]
	nCoV_IP4-14146Rv	CTGGTCAAGGTTAATATAGG		
	2019-nCoV_N2-F	TTACAAACATTGGCCGCAAA	Gene N	[9]
	2019-nCoV_N2-R	GCGCGACATTCCGAAGAA		
	Gen_N-105F	TCATCACGTAGTCGCAACAG	Gene N	This study
	Gen_N-105R	CAAAGCAAGAGCAGCATCAC		
	Gen_N-90F	GGCAGTAACCAGAATGGAGAA	Gene N	This study
	Gen_N-90R	GGTGAACCAAGACGCAGTAT		
	Gen_E-107F	ATTCGTTTCGGAAGAGACAGG	Gene E	This study
	Gen_E-107R	ATCGAAGCGCAGTAAGGATG		
RT-LAMP	E1-LF	CGCTATTAACTATTAACG	Gene E	[10]
	E1-LB	GCGCTTCGATTGTGTGCGT		
	E1-F3	TGAGTACGAACTTATGTACTCAT		
	E1-B3	TTCAGATTTTTAACACGAGAGT		
	E1-FIP	ACCACGAAAGCAAGAAAAAGAAGTTTTTTCGTTTCGGAAGAGACAG		
	E1-BIP	TTGCTAGTTACACTAGCCATCCTTACTTTTGTTTTACAAG ACTCACGT		
	N2-LF	GGGGGCAAATTGTGCAATTTG	Gene N	[10]
	N2-LB	CTTCGGGAACGTGGTTGACC		
	N2-F3	ACCAGGAACTAATCAGACAAG		
	N2-B3	GACTTGATCTTTGAAATTTGGATCT		
	N2-FIP	TTCCGAAGAACGCTGAAGCGTTTTAACTGATTACAAACATTGGCC		
	N2-BIP	CGCATTGGCATGGAAGTCACAATTTTTTTGATGGCACCTGTGTA		

NA—not applicable.

**Table 3 biomedicines-10-02329-t003:** Limit of detection (LOD) for RT-HDA combined with LFA using a synthetic template.

Concentration of the Synthetic Template (µM)	Incubation Time (Min)						
	90	75	60	45	30	20	10
0.2	+	+	+	+	+	+	+
0.02	+	+	+	+	+	(+) *	(+)
0.002	+	+	+	+	-	-	-
0.0002	(+)	(+)	-	-	-	-	-
Negative control	-	-	-	-	-	-	-

* weak reaction is shown in brackets.

**Table 4 biomedicines-10-02329-t004:** Limit of detection (LOD) for RT-HDA combined with LFA using control RNA as a template.

Concentration of the Synthetic Template (Copies/Reaction)	Incubation Time (Min)				
	60	45	30	20	10
30000	+	+	+	+	+
3000	+	+	+	+	(+)*
300	+	+	+	+	(+)*
30	+	+	+	+	(+)*
3	+	+	+	+	(+)*
Negative control	-	-	-	-	-

* weak reaction is shown in brackets.

**Table 5 biomedicines-10-02329-t005:** Limit of detection for RT-LAMP combined with LFA using control RNA as a template.

Concentration of the Template (Copies/Reaction)	Incubation Time (Min)				
	60	50	40	30	20
30000	+	+	+	+	+
3000	+	+	+	+	-
300	+	+	+	-	-
30	+	+	(+) *	-	-
3	-	-	-	-	-
Negative control	+	+	-	-	-

* weak reaction is shown in brackets.

**Table 6 biomedicines-10-02329-t006:** Summary of LFA based tests for COVID-19 diagnostics [2].

Type of the Test	LOD	Sensitivity	Specificity	Time
Antibody -targeting COVID-19 tests	100 pf/mL–1.2 mg/mL	44%–100%	78%–100%	5 min–60 min
Nucleic acid -targeting SARS-CoV-2 tests	0.02 copies/reactions–2300 copies/reactions	82%–100%	82%–100%	<10 min–90 min
Antigen–targeting SARS-CoV-2 tests	0.016 fg/mL–2.2 ng/mL	69%–93%	93%–100%	5 min–20 min

## Data Availability

Not applicable.

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
