# Peer review of "Detection of SARS-CoV-2 Using Reverse Transcription Helicase Dependent Amplification and Reverse Transcription Loop-Mediated Amplification Combined with Lateral Flow Assay"

_biomedicines, 2022, doi:10.3390/biomedicines10092329_

Round 1

Reviewer 1 Report

The authors present two isothermal amplification methods, HDA and LAMP, combined with a commercially available strip / lateral flow assay for the detection of E and N gene of SARS-CoV-2. However, LAMP, as well as RPA (similar to HAD) methods have already been reported, two of which have also used the same available strips (Zhu et al. 2022, Biosens Bioelectron 166, 112437 doi.org/10.1016/j.bios.2020.112437; Chen et al. 2021 Frontiers in Cell Infection Microbiol doi: 10.3389/fcimb.2021.581239; Agarwal et al, 2022 Anal Bioanal Chem 414:3177–3186 doi.org/10.1007/s00216-022-03880-4; Zhang et al 2021, Anal Chem 93, 3325−3330 dx.doi.org/10.1021/acs.analchem.0c05059; Lau et al 2021, PLoS ONE 16(1): e0245164. https://doi.org/10.1371/journal.pone.0245164).

The authors do not present clearly the advantages of their method and what is the novelty of their method. HDA has not be used for the isothermal amplification of SARS-CoV-2, but the authors should present the advantages over the other amplification techniques. An overview and a comparison with other available lateral flow assays should also be included. Taking into consideration the following issues, I cannot suggest publication of this manuscript in its present form.

 -        In Figure 3, for the 10 min incubation, I observe that there is no increase of the signal of the test zone with increasing of the target concentration up to 1000 times!! Same is observed at 20 min. This may be attributed to contamination. The signal of the test zone should be increased with the increased concentration after 100 or 1000 times more RNA copies. The authors should perform again this experiment with fresh aliquots of the reagents.

 -        Please add to which gene AMPLI-RUN SARS-CoV-2 RNA is refer to. Which primers are used for its amplification?

 -        Samples have not be tested with the lateral flow assay. Real sample analysis has to be included.

 -        Specificity and Reproducibility has to be determined for both methods.

 -        The authors state that a multiplex amplification reaction for E and N gene. However, both genes are detected with the same manner with the strip and no distinction is made. A multiplex reaction, though may lower the detectability than a single amplification reaction. The authors have to investigate this.

 -        As for the false positive, are the authors sure that this is not a contamination? Again here, I have this concern due to the results in figure 3.

 -        Calibration graphs with more target concentration is also needed for RT-LAMP method.

Author Response

Dear Reviewer,

Thank you very much for the review which is very helpful for us to improve the manuscript. Below we present our responses to each of the comments.

The authors present two isothermal amplification methods, HDA and LAMP, combined with a commercially available strip / lateral flow assay for the detection of E and N gene of SARS-CoV-2. However, LAMP, as well as RPA (similar to HDA) methods have already been reported, two of which have also used the same available strips (Zhu et al. 2022, Biosens Bioelectron 166, 112437 doi.org/10.1016/j.bios.2020.112437; Chen et al. 2021 Frontiers in Cell Infection Microbiol doi: 10.3389/fcimb.2021.581239; Agarwal et al, 2022 Anal Bioanal Chem 414:3177–3186 doi.org/10.1007/s00216-022-03880-4; Zhang et al 2021, Anal Chem 93, 3325−3330 dx.doi.org/10.1021/acs.analchem.0c05059; Lau et al 2021, PLoS ONE 16(1): e0245164. https://doi.org/10.1371/journal.pone.0245164).

We agree that RPA is somewhat similar to HDA. However, there are also important differences between the two methods, for example the RPA reaction mixture is more complex and contains, among others, appropriate recombinase and polymerase and various proteins such as single-stranded DNA binding protein, recombination mediator protein, as well as additional substances like Carbowax20M, as a crowding agent (Piepenburg et al. PLoS Biol. 2006 Jul;4(7):e204. doi: 10.1371/journal.pbio.0040204). RPA enables amplification of fragments in size up to 800 bp at 37°C but due to the complex reaction mixture the product must be purified before the detection step, for example before agarose gel electrophoresis (TwistDx. TwistAmp DNA Amplification Kits. Combined Instruction Manual). The HDA reaction mixture contains, among others, helicase and polymerase as well as auxiliary proteins (Vincent et al. EMBO Rep. 2004 Aug;5(8):795-800. doi: 10.1038/sj.embor.7400200). HDA enables amplification of fragments in size up to 130 bp at 65°C but no additional purification step is necessary before the product detection. In our opinion the differences between these two methods are crucial for application in diagnostic tests.

We have mentioned in the manuscript that LAMP has been used for SARS-CoV-2 detection by many researchers (Discussion section, page 9: “Among various isothermal amplification methods, the RT-LAMP is the most widely used for the development of COVID-19 rapid tests…”). In our study we applied LAMP for comparison to HDA as LAMP is a broadly used isothermal amplification method and might be regarded as a reference among nucleic acid isothermal amplification methods. We believe that comparison of two methods in one laboratory by the same staff, in the same environment and using the same samples and reagents, where possible, is much more reliable than comparison of a developed method only to published results obtained in another laboratory and under another conditions.

The authors do not present clearly the advantages of their method and what is the novelty of their method. HDA has not be used for the isothermal amplification of SARS-CoV-2, but the authors should present the advantages over the other amplification techniques. An overview and a comparison with other available lateral flow assays should also be included.

The advantages of RT-HDA over the RT-LAMP are described in Conclusions and include higher sensitivity, minimization of the risk of false positive results caused by primers interactions and possibility of application of primer pairs recommended by the WHO which enables to monitor suitability of the test for the diagnosis of emerging new variants of the virus. We have added description of the advantages of RT-HDA and RT-LAMP over the antibody and antigen LFA tests as well as nucleic acid amplification tests combined with colorimetric detection in the Discussion section.

 -        In Figure 3, for the 10 min incubation, I observe that there is no increase of the signal of the test zone with increasing of the target concentration up to 1000 times!! Same is observed at 20 min. This may be attributed to contamination. The signal of the test zone should be increased with the increased concentration after 100 or 1000 times more RNA copies. The authors should perform again this experiment with fresh aliquots of the reagents.

In Figure 3, for the 10 min incubation a strong test band is visible for the template concentration 0.1 µM and a weak test band for the concentration 0.01 µM. For the 20 min incubation a strong test band is visible for the template concentration 0.1 µM, a weak test band is visible for the concentration 0.01 µM, and for the template concentration 0.001 µM there is no test band visible. Therefore, the Reviewer’s remark is not clear to us.

We excluded the contamination because for each reaction mixture and incubation time a negative control was added and the control was always correct. All the tests were performed by at least two laboratory workers and repeated at least three times. We have added this information to the Materials and Methods section, point 2.9.

 -        Please add to which gene AMPLI-RUN SARS-CoV-2 RNA is refer to. Which primers are used for its amplification?

The AMPLIRUN SARS-CoV-2 RNA is the purified total RNA from cultivated SARS-CoV-2 clinical isolates as we have mentioned in the section Materials and Methods, point 2.1. It have been used for amplification with all the primers described in the manuscript.

 -        Samples have not be tested with the lateral flow assay. Real sample analysis has to be included.

Information concerning the analysis of clinical samples has been added in the sections Materials and Methods and Results.

 -        Specificity and Reproducibility has to be determined for both methods.

The information concerning specificity and reproducibility has been in the sections Materials and Methods and Results.

 -        The authors state that a multiplex amplification reaction for E and N gene. However, both genes are detected with the same manner with the strip and no distinction is made. A multiplex reaction, though may lower the detectability than a single amplification reaction. The authors have to investigate this.

The both genes were detected with the same manner and the duplex format has not influenced on the LOD. We have added these information in the Discussion section.

 -        As for the false positive, are the authors sure that this is not a contamination? Again here, I have this concern due to the results in figure 3.

 We excluded the contamination because for each reaction mixture and incubation time a negative control was added and the control was always correct. All the tests were performed by at least two laboratory workers and repeated at least three times.

 -        Calibration graphs with more target concentration is also needed for RT-LAMP method.

The Reviewer’s remark is not clear for us. The RT-LAMP combined with LFA is a qualitative not a quantitative method. Calibration graphs are constructed usually for quantitative methods. Various target concentrations we have used to establish he limit of detection at different incubation times, what is presented in Table 5.

Reviewer 2 Report

This manuscript reported the SARS-CoV-2 detection via RT-HAD and RT-LAMP on test strip. It is a continuously interesting topic and deserved to investigate. The experiments were well designed and constructed and supported the results.

Major

1.      Introduction. The current profile of SARS-CoV-2 detection should be discussed while this introduction could not cover the cutting-edge information.

2.      Please state clearly what is the novelty compared to existing papers based on test strip via antibody, and in-tube test via PCR, LAMP, RCA and so on.

3.      Please add the sample preparation process in both experiment and result.

4.      Please add the results of interference, inner-assay, inter-assay, sensitivity, and specificity.

Author Response

Dear Reviewer,

Thank you very much for the review which is very helpful for us to improve the manuscript. Below we present our responses to each of the comments.

  1. The current profile of SARS-CoV-2 detection should be discussed while this introduction could not cover the cutting-edge information.

We have added information about detection methods including CRISPR-Cas, biosensors and sequencing.

  1. Please state clearly what is the novelty compared to existing papers based on test strip via antibody, and in-tube test via PCR, LAMP, RCA and so on.

We have discussed  advantages of RT-LAMP and RT-HDA combined with LFA in comparison to antibody and antigen LFA, as well as nucleic acid amplification methods combined with colorimetric in-tube detection in the Discussion section.

  1. Please add the sample preparation process in both experiment and result.

The information has been added in the Materials and Methods section.

  1. Please add the results of interference, inner-assay, inter-assay, sensitivity, and specificity.

We have added the description of investigation of sensitivity, specificity, accuracy and reproducibility in the sections Materials and Methods and Results.

In the process of improving the manuscript we have added a few references. The manuscript has been corrected by the language proofreading service: ArcusLink (https://www.arcuslink.pl/en/)

Round 2

Reviewer 1 Report

The authors have responded to most of the Reviewers' suggestions.

An overview and a comparison with other available lateral flow assays is still needed. The comparison can be added in a form of a Table, including limit of detection, analysis time, specificity, reproducibility, myltiplexicity and real sample analysis.

Author Response

Dear Reviewer,

Thank you for the suggestion. We have included the table showing summary of LFA based tests apply for COVID-19 diagnostics. We have decided to present a summary table instead of detailing each test featured in the scientific literature because there are so many LFA tests described in scientific papers that a separate review article should be written.

Kind regards,

Reviewer 2 Report

This manuscript was revised well and can be published.

Author Response

Dear Reviewer,

Thenk you very much for your help.

Kind regards,